# Assessment of Human Settlement Quality Based on the Population Exposure Risk to PM$_{2.5}$ Pollution in the Mid-Spine Belt of Beautiful China

Qiang Yang [1,*], Juncheng Fan [1], Jie Min [1], Jiaming Na [1], Pengling Wang [2], Xinyuan Wang [3,4], Ruichun Chang [4] and Quanfeng Wang [4]

1   College of Civil Engineering, Nanjing Forestry University, Nanjing 210037, China; fanjuncheng@njfu.edu.cn (J.F.); jiemin@njfu.edu.cn (J.M.); jiaming.na@njfu.edu.cn (J.N.)
2   National Climate Centre, China Meteorological Administration, Beijing 100081, China; wangpl@cma.gov.cn
3   Key Laboratory of Digital Earth Science, Institute of Remote Sensing and Digital Earth (RADI), Chinese Academy of Sciences (CAS), Beijing 100094, China; wangxy@aircas.ac.cn
4   Digital Hu Line Research Institute, Chengdu University of Technology, Chengdu 610059, China; changruichun08@cdut.edu.cn (R.C.); wangquanfeng@cdut.edu.cn (Q.W.)
*   Correspondence: qiangyang@njfu.edu.cn

**Abstract:** Human settlements are areas closely related to human production and life, and their quality directly affects people's physical health and quality of life. However, with air pollution continuing to worsen, people are becoming increasingly concerned about the exposure to air pollution in their residential regions. In addition, many studies ignore the long-term impact of environmental factors on the evolution of human settlement quality (HSQ). This study aims to assess the HSQ in the Mid-Spine Belt of Beautiful China (MSBBC) from 2000 to 2020 and to analyze the changes in its driving forces during different time periods. We divided the HSQ into five dimensions: terrain conditions, climate conditions, hydrological conditions, ground cover conditions, and air pollution exposure. The Entropy-TOPSIS and grey relational degree model were used to assess the HSQ in the MSBBC. To analyze the driving forces of HSQ, the optimal-parameters-based geographic detector model was utilized. The results show the following: (1) Within the study area, the degree of the population exposure risk to PM$_{2.5}$ and its change trend were significantly different on both sides of the Hu Line, with the east showing higher levels than the west. (2) The HSQ of the MSBBC decreased from east to west. Moreover, the HSQ in the metropolitan area of the urban agglomeration was characterized from low to high. The HSQ underwent three phases, consisting of an increase (2000–2010), a decrease (2010–2015), and an increase again (2015–2020). (3) Among the driving factors, the impact of PM$_{2.5}$ on the HSQ weakened year by year. The driving force of meteorological parameters on the HSQ was the strongest. Furthermore, the interactions of each factor could enhance the HSQ. The results of this study not only provide a strong reference for creating livable areas and promoting sustainability in the MSBBC but also contribute to addressing challenges such as pollution and climate change.

**Keywords:** human settlement quality (HSQ); population exposure risk to PM$_{2.5}$; Mid-Spine Belt of Beautiful China (MSBBC); OPGD model; Entropy-TOPSIS and grey relational degree model





## 1. Introduction

Human settlements are not only the spaces where human production, work, and consumption take place but also an important determinant of the long-term and stable development of society, economy, and environment within each country [1,2]. Since the first industrial revolution, great changes have occurred in the global environment (air pollution, ecological degradation, heat island effect), resulting in dramatic changes in the environmental quality of human residential areas, especially in emerging economy countries [3,4]. Over the past forty years, due to the rapid growth of China's economy,

ecological vulnerability, air and water pollution, and other issues have become increasingly prominent [5]. Fortunately, the Chinese government has gradually become aware of the importance of environmental protection, and has started to regulate various behaviors that pollute the environment [6,7]. This typical phenomenon of "pollution before treatment" can cause dramatic changes in the ecosystem, directly feeding back to the human settlement quality (HSQ), which closely affects us [8,9]. The issue of HSQ management has become a focus of common concern for governments and society [6,9]. These challenges not only have a profound impact on human health and quality of life but also pose a huge threat to the global society, economy, and ecosystem [10]. The quality and sustainability of human settlements have become one of the most pressing issues of our time, affecting the daily lives of billions of people and the fate of future generations [10,11]. Therefore, a deep understanding of the past, present, and future development status of HSQ is key to solving this problem.

Since the concept of HSQ science was proposed in the 1960s [12], scholars from different fields have discussed the research topic and formed different perspectives: (1) Objective analysis of living and supporting systems in HSQ [1,3,13]; (2) Subjective sentiment analysis based on questionnaires and interviews [14–16]; (3) Assessment of the HSQ using a single element of a natural system or a few elements [11]. These studies have deepened people's understanding of HSQ in many aspects. First, they reveal to people the complexity and scale uncertainty of the HSQ. Second, the study of HSQ has gradually developed a methodology and promoted the development of other subsidiary disciplines. Finally, some of the natural and environmental factors affecting the HSQ have been revealed and confirmed by these studies. Although related scholars have conducted detailed studies on the risk factors of HSQ, they often neglect the importance of population-based air pollution exposure levels, which is a decisive factor of health risks in HSQ [17,18]. Among them, atmospheric $PM_{2.5}$ (i.e., fine particulate matter with an aerodynamic diameter $\leq 2.5$ um) pollution has become the most serious threat to human health [19]. For example, research has shown that long-term exposure to $PM_{2.5}$ pollution may cause cardiovascular diseases such as lung cancer, myocardial infarction, and myocardial ischemia, as well as be acute triggers of common respiratory diseases such as asthma, bronchitis, rhinitis, and upper and lower respiratory tract infections [20]. The World Health Organization (WHO) emphasizes that $PM_{2.5}$ is a major global risk factor for air pollution: $PM_{2.5}$ has caused about 4.14 million premature deaths worldwide each year due to chronic exposure to heavily polluted air, and China accounts for 30% of them, ranking first in the world [19,20]. In addition, $PM_{2.5}$ is also a key factor leading to the distinctive two-level differentiation of the HSQ in Chinese cities, seriously affecting the HSQ in densely populated areas [21]. As a common problem of global urban air pollution, the impact of the population exposure risk of $PM_{2.5}$ (PER-$PM_{2.5}$) on the HSQ must be studied.

In addition, the assessment of HSQ must be improved. On the one hand, most dimensions of HSQ evaluation mainly focus on the characteristics of the study area, dividing it into different spatial units, and then analyzing the spatial changes of HSQ [22]. However, in the spatial dimension, only several aspects such as the economic development, climate, and urban morphology of HSQ are displayed, which does not easily reflect the continuous causal relationship of the HSQ [23]. For example, Hu et al. used statistical yearbook data from a certain year to spatially classify the development potential of HSQ in rural China, but they did not consider the causal relationship between the HSQ in each region at different time periods [23]. Taking into account both the temporal and spatial dimensions of HSQ is fundamental to formulating targeted governance strategies. On the other hand, although the existing research data sources are relatively rich, few comprehensively evaluate HSQ on the basis of high-resolution grid data. At the same time, most studies use statistical yearbook data and questionnaire survey data. These two data sources are not only difficult to obtain but also have limited data volume, making it difficult to accurately reflect the continuous spatial changes in the living environment [24,25]. These limitations make the existing data only applicable to macro research on HSQ [14]. While some studies have

attempted to use GIS spatial interpolation technology to estimate survey data in order to compensate for the shortcomings of yearbook and questionnaire data to some extent, this method has significant limitations that cannot be ignored [26]. First, it ignores the non-uniformity of data distribution, which may lead to biased research results. Second, the spatial resolution of this method is usually low and cannot accurately capture complex and subtle changes in geographic space. Therefore, using high-spatial-resolution grid data has become an indispensable means. This type of data not only provides more detailed spatial analysis but also more accurately captures micro changes in the quality of the HSQ [22]. This is of decisive significance for conducting a comprehensive and accurate evaluation of regional HSQ [3,17].

The construction of the Mid-Spine Belt of Beautiful China (MSBBC) is a strategic requirement for the new planning of national land space under the new era of high-quality development models, and it is of great significance for building new urbanization demonstration areas and new livable living areas [27]. Therefore, this study selected the MSBBC as the research area and used high-resolution satellite data to reveal a large degree of PER-PM$_{2.5}$ locally in the MSBBC by combining the impact of industrial transformation and clean-air policies [27]. This was used as an indicator to construct an improved habitat evaluation model to analyze its spatial–temporal characteristics and driving factors. The main objectives of this study are as follows: (1) To reveal the spatial–temporal evolution characteristics of the PER-PM$_{2.5}$ in the MSBBC from 2000 to 2020; (2) To measure changes in the HSQ in the MSBBC from 2000 to 2020; (3) To investigate the driving factors of HSQ via the optimal-parameters-based geographic detector (OPGD) model.

## 2. Materials and Methods

### 2.1. Study Area

The MSBBC is in the mid-spine position of China's territory (45° running through the northeastern and southwestern regions of China), between 23°36′–50°57′ N and 97°25′–133°21′ E. The total area of the MSBBC is about 2.444 million km$^2$, accounting for 25.4% of China's land area (Figure 1). It is an envelope zone formed by the displacement of the abrupt change line of China's population density (Hu Line) over the past 80 years [28]. Its natural and cultural background is that of an agro-pasture ecotone with a fragile ecology [28]. The terrain of the area is intertwined and complex, including five basic landforms: plateau, mountain, hill, plain, and basin. The terrain fluctuates greatly, mainly showing a low northeast, high southwest trend [29].

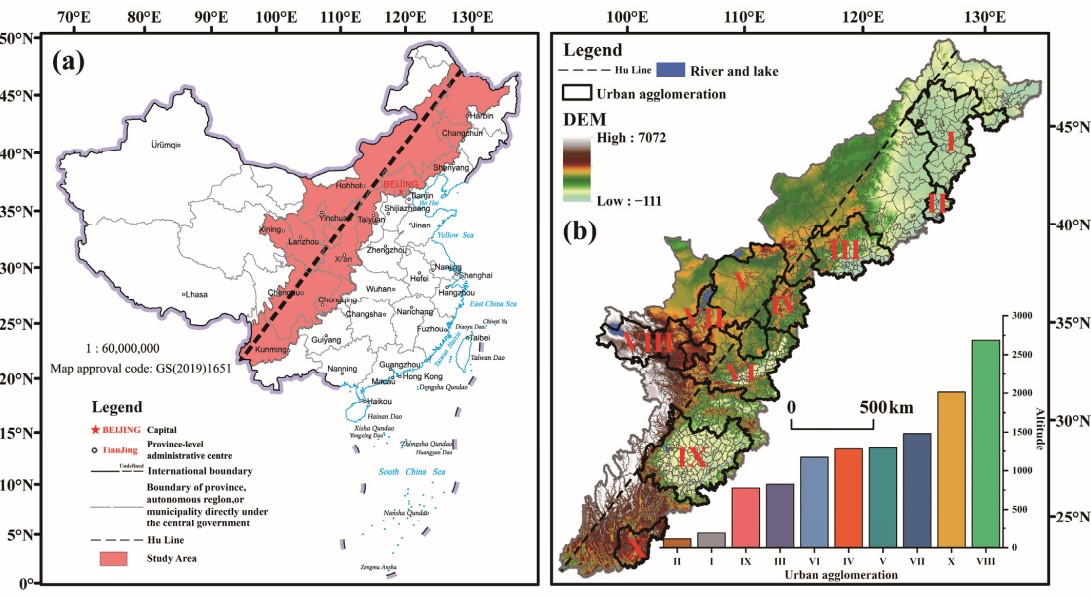

**Figure 1.** The location (**a**) and elevation (**b**) of the study area.

According to the National New Urbanization Plan (2014–2020), 19 urban agglomerations will be steadily built in the future [30], and the MSBBC includes more than 10 of these (Table 1). With the continuous promotion of urban agglomeration construction, the economy, ecology, and culture of the MSBBC have significantly improved. Meanwhile, due to the different development models of different regions on both sides of the Hu Line, the gap in the quality of the living environment will further widen for regions that have long pursued industrial construction and neglected environmental protection [31]. Therefore, evaluating the HSQ in the MSBBC will provide a reference for reducing the imbalance in its development and is an important channel to solve the scientific problem of the Hu Line.

**Table 1.** Urban agglomerations involved in the MSBBC.

| Urban Agglomeration | Abbreviation * |
| --- | --- |
| Harbin–Changchun urban agglomeration | I |
| Central and Southern Liaoning Province urban agglomeration | II |
| Beijing–Tianjin–Hebei urban agglomeration | III |
| Jinzhong urban agglomeration | IV |
| Hohhot–Baotou–Ordos–Yulin urban agglomeration | V |
| Guanzhong Plain urban agglomeration | VI |
| Urban belt along the Yellow River in Ningxia | VII |
| Lanxi urban agglomeration | VIII |
| Chengdu–Chongqing urban agglomeration | IX |
| Yunnan central urban agglomeration | X |

* Note: All the urban agglomerations mentioned below are referred to by their abbreviations.

### 2.2. Construction of the Index System

This study studies human settlements from a natural perspective. Considering the inherent characteristics of the living environment of the MSBBC, the factors that have a major impact on the MSBBC are selected as the evaluation factors for this study. Although there are many natural factors that affect the HSQ, the most fundamental and dominant factors for the natural suitability of human environments mainly include terrain conditions, climate conditions, hydrological conditions, and ground cover conditions [11,18]. In addition, PER-PM$_{2.5}$, as one of the natural environmental factors, has rarely been studied to elucidate its role in the human settlement environment. In this paper, according to the Technical Specifications for Ecological Environment Assessment (HJ192-2015) issued by the Ministry of Environmental Protection of the People's Republic of China [32], the natural environment element indicators are selected, including the hydrological index, topographic relief, ground cover index, and temperature–humidity index [10,33]. At the same time, PER-PM$_{2.5}$ is considered as one of the quality indicators of the living environment.

In recent years, with the development of 3S technology and the deepening of research on the relationship between the regional population, resources, the environment, and development, many scholars at home and abroad have taken topographic relief as an important indicator for regional resource environment evaluation and regional resource environment carrying-capacity evaluation [34,35]. It has been widely applied in areas such as regional geological environment evaluation, ecological environment condition evaluation, natural disaster evaluation, regional sustainable development evaluation, urban land grading and grading, and human settlement environment suitability evaluation [34]. The temperature–humidity index is a biological meteorological indicator that evaluates the comfort level of the human body under different climatic conditions from a meteorological perspective, based on the principle of heat exchange between the human body and external meteorological elements [36]. It effectively reflects the human body's perception of the degree of environmental cold and heat and has a significant impact on the improvement in the quality of the living environment [13]. Previous studies have shown that water resources are one of the important natural factors affecting population distribution and the living environment, and they play an indispensable role in maintaining the regional basic living environment [37,38]. Therefore, when adjusting and organizing the population

layout and improving the regional living environment, it is necessary to fully consider water resource factors [39]. Vegetation plays an important role in the climate, hydrology, and biochemical cycles by affecting the energy balance of the earth's atmosphere system [11,33]. It is a sensitive indicator of the impact of climate and human factors on the environment. To some extent, the state of vegetation coverage can be said to be an indicator of the quality of the living environment [40].

### 2.3. Data Sources and Preprocessing

The $PM_{2.5}$ concentration data used in this study are from the China High $PM_{2.5}$ dataset in China High Air Pollution. The cross-validation showed that the annual $PM_{2.5}$ estimates at the verification point are highly consistent with the measured values ($R^2 = 0.91$ and RMSE = 5.07 μg/m$^3$) [41]. Therefore, using this data can better simulate the changes in $PM_{2.5}$ concentration at different spatial–temporal scales with high accuracy [42]. This dataset covers the period from 2000 to 2020 with a spatial resolution of 1 km. The population data are derived from the LandScan global population grid data with a spatial resolution of approximately 1 km [43].

The spatial resolution of the meteorological parameters (temperature, precipitation) and relative humidity is about 1 km, and these were averaged every five years as 2000, 2005, 2010, 2015, and 2020. The detailed data description and source are shown in Table 2. All the above data were converted into raster data with a spatial resolution of 1 km.

**Table 2.** HSQ System and Data Source of the MSBBC.

| Category | Data | Period | Source |
|---|---|---|---|
| PER-$PM_{2.5}$ | $PM_{2.5}$ | 2000–2020 | China High Air Pollution (https://weijing-rs.github.io/product.html) (accessed on 5 November 2022) |
| | population | | LandScan (https://landscan.ornl.gov) (accessed on 28 November 2022) |
| topographic relief | DEM | 2009 | Geospatial Data Cloud (http://www.gscloud.cn) (accessed on 25 October 2022) |
| temperature–humidity index | temperature | 1996–2020 | National Earth System Science Data Center (http://www.geodata.cn) (accessed on 2 December 2022) |
| | relative humidity | | |
| hydrological index | precipitation | | |
| | water area | 2000, 2005, 2010, 2015, 2020 | |
| | land use | | |
| ground cover index | NDVI | 2000–2020 | National Ecosystem Science Data Center (http://www.nesdc.org.cn) (accessed on 12 November 2022) |

### 2.4. Methods

The technical roadmap for the assessment of HSQ and the driving force analysis (Figure 2) and its specific steps are as follows:

(1) Relevant data were collected and then converted to grid data; afterward, the projection coordinate system and grid size were set. A total of nine evaluation index factors were selected from the aspects of hydrological index, topographic relief, ground cover index, temperature–humidity index, and PER-$PM_{2.5}$ to establish an evaluation index system.

(2) Time-series analysis of PER-$PM_{2.5}$ was performed using the Theil–Sen median, the Mann–Kendall (MK), and the Hurst index methods.

(3) The nine indicator factors were normalized, and the weights of each index were determined by TOPSIS and the grey relational degree model. Then, the corresponding models were used to build HSQ models for 2000, 2005, 2010, 2015, and 2020.

(4) The OPGD model was used to detect the driving factors of HSQ.

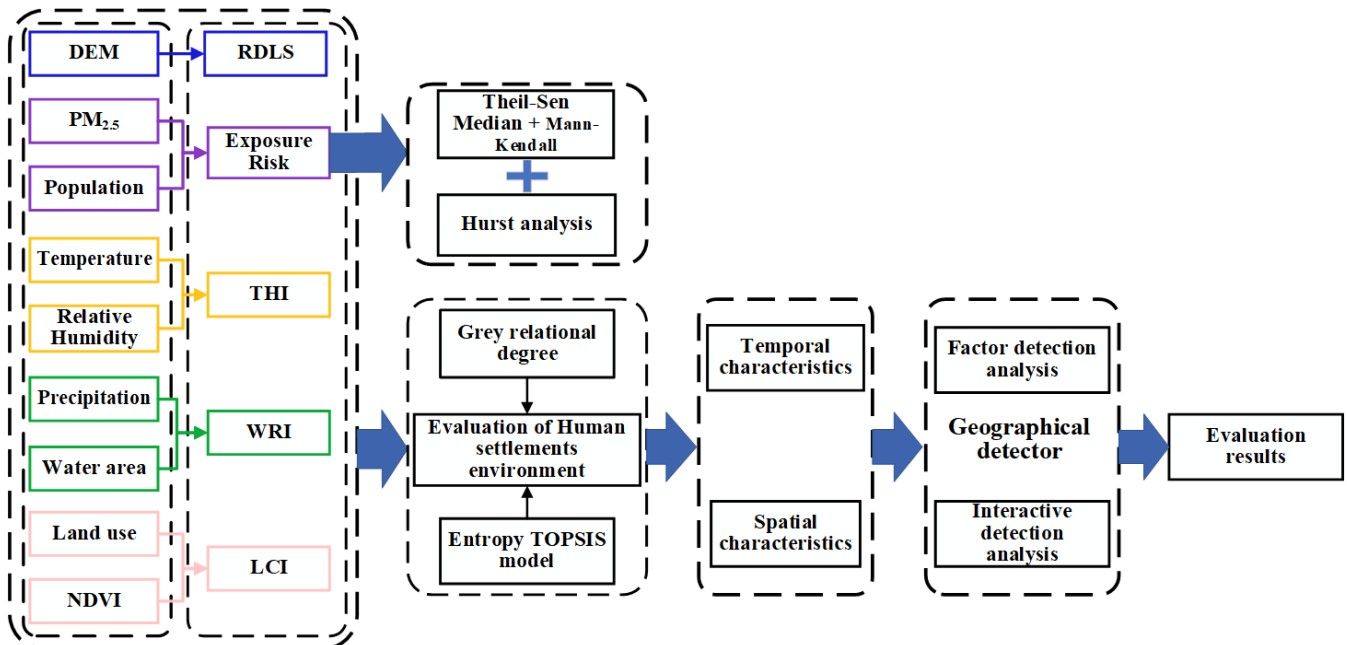

**Figure 2.** Flowchart of this study.

2.4.1. Entropy-TOPSIS and Grey Relational Degree Model

The Entropy-TOPSIS and grey relational degree model is an evaluation algorithm that obtains the weights of various indicators through the entropy method and fuses the grey correlation model on the basis of TOPSIS method [44–46]. This model avoids the subjective factor interference of the traditional TOPSIS method, which uses a subjective weighting method to calculate weights, and can objectively reflect the dynamic and changing trends of the evaluation object [45]. Moreover, this method can evaluate the merits of schemes in two dimensions: static distance and dynamic trend, overcoming the shortcomings of a single traditional model that does not fully utilize existing information [47]. The study assumed that m samples were evaluated, including *n* evaluation indexes; the corresponding index values were $x_{ij}$ ($i$ = 1, 2, ..., $m$; $j$ = 1, 2, ..., $n$), and the decision matrix was $X = \{x_{ij}\}_{m \times n}$.

(1) Weighted standardization of the index matrix

    (a) Standardize the indicator matrix: Because of the differences in the units and quantities of the original evaluation indicators, the data need to be standardized for their comparative analysis [48]. For the standardized decision matrix $Y = \{y_{ij}\}_{m \times n}$, the calculation formula is as follows: When larger values are better, the standardized data $y_{ij}$ is:

$$y_{ij} = \frac{x_{ij} - \min(x_{ij})}{\max(x_{ij}) - \min(x_{ij})} \tag{1}$$

When smaller values are better, the normalized data $y_{ij}$ is:

$$y_{ij} = \frac{\max(x_{ij}) - x_{ij}}{\max(x_{ij}) - \min(x_{ij})} \tag{2}$$

(b)     Homogenize the indicators, and calculate the proportion $p_{ij}$ of the *i*-th grid unit under the *j*-th HSQ indicator:

$$q_{ij} = \frac{y_{ij}}{\sum\limits_{i=1}^{m} y_{ij}} \tag{3}$$

(c)     Calculate the information entropy $e_j$:

$$E_j = -\frac{1}{\ln n}\sum_{i=1}^{m} q_{ij} \ln q_{ij} \tag{4}$$

(d)     Calculate the corresponding comprehensive weight $W_j$ for each indicator:

$$W_j = \frac{1 - E_j}{\sum\limits_{j=1}^{n} (1 - E_j)} \tag{5}$$

(2)   TOPSIS–grey correlation degree model TOPSIS is a method of ranking the strengths and weaknesses of multiple evaluation objects by approximating the Euclidean distance between multiple evaluation objects and the ideal solution, and then correlating the strengths and weaknesses of multiple evaluation objects [49]. However, the method uses Euclidean distance to measure the positional relationship between the indicators, which makes it impossible to rank their positions. In this study, the grey relational degree was used to improve the Euclidean distance [47], which made the calculation results more accurate and reasonable.

(a)     Construct a weighted normalization matrix Z based on standardized indicators and their corresponding weights:

$$(Z_{ij})_{m\times n} = y_{ij} \times W_j \left(i = 1, 2, \cdots, m \,; j = 1, 2, \cdots, n\right) \tag{6}$$

(b)     Determine the positive and negative ideal solution of the evaluation object based on its maximum and minimum values:

$$Z^+ = (z_1^+, z_2^+ +, \cdots, z_j^+), \ Z^- = (z_1^-, z_2^-, \cdots, z_j^-) \tag{7}$$

where $z_j^+ = (z_{ij})_{\max}$, $z_j^- = (z_{ij})_{\min}$.

(c)     Determine the distance $d_i^+$ and $d_i^-$ from the *i*-th grid to the positive and negative ideal solution:

$$d_i^+ = \sqrt{\sum_{j=1}^{n} (z_{ij} - z_j^+)^2}, \ d_i^- = \sqrt{\sum_{j=1}^{n} (z_{ij} - z_j^-)^2} \tag{8}$$

(d)     Calculate the index value of each grid and the grey correlation coefficient matrices $R^+$ and $R^-$ of the positive and negative ideal solutions $z_j^+$ and $z_j^-$, respectively, for the *j*-th index:

$$\begin{aligned} R^+ &= (r_{ij}^+)_{m\times n}, \ R^- = (r_{ij}^-)_{m\times n} \\ r_{ij}^+ &= \frac{\min(|z_j^+ - z_{ij}|) + \rho \times \max(|z_j^+ - z_{ij}|)}{|z_j^+ - z_{ij}| + \rho \times \max(|z_j^+ - z_{ij}|)} \\ r_{ij}^- &= \frac{\min(|z_j^- - z_{ij}|) + \rho \times \max(|z_j^- - z_{ij}|)}{|z_j^- - z_{ij}| + \rho \times \max(|z_j^- - z_{ij}|)} \end{aligned} \tag{9}$$

where $\rho \in (0, 1)$ is the resolution factor. In accordance with the available literature, this value is taken as 0.5. Therefore, the grey correlation degrees $r_i^+$ and $r_i^-$ of the *i*-th grid are as follows:

$$r_i^+ = \frac{1}{n}\sum_{j=1}^{n} r_{ij}^+, \; r_i^- = \frac{1}{n}\sum_{j=1}^{n} r_{ij}^- \tag{10}$$

(e)   Based on the above results, the determined distance and correlation degree were processed by a dimensionless method:

$$D_i^+ = \frac{d_i^+}{\max(d_i^+)}, \; D_i^- = \frac{d_i^-}{\max(d_i^-)}$$
$$R_i^+ = \frac{r_i^+}{\max(r_i^+)}, \; R_i^- = \frac{r_i^-}{\max(r_i^-)} \tag{11}$$

(f)   In summary, when the values of $D_i^-$ and $R_i^+$ are larger, the evaluation object is closer to the ideal scheme; conversely, when the values of $D_i^-$ and $R_i^+$ are smaller, the more distant the evaluation object is from the ideal scheme. Therefore, the above equations are combined:

$$T_i^+ = \frac{D_i^- + R_i^+}{2}, \; T_i^- = \frac{D_i^+ + R_i^-}{2}$$
$$S_i = \frac{T_i^+}{T_i^+ + T_i^-} \tag{12}$$

where $T_i^+$ and $T_i^-$ reflect the distance between the evaluation object and the ideal value from the positive and negative sides, respectively. $S_i$ is the grey correlation closeness degree. The greater the closeness degree, the better the HSQ; on the contrary, the smaller the closeness degree, the poorer the HSQ.

2.4.2. Indicator Selection Basis and Construction of the HSQ Model

(1)   Population exposure risk assessment model of PM2.5

To quantify the risk degree of population exposure risk to $PM_{2.5}$ pollution in the region, this study introduced the relative risk assessment model for population air pollution exposure [50]:

$$PRE - PM_{2.5}^i = \frac{P_i \times C_i}{\sum\limits_{i=1}^{n} P_i \times \frac{C_i}{n}} \tag{13}$$

where $PRE\text{-}PM_{2.5}^i$ is the relative population exposure level, *i* denotes each pixel, $P_i$ is the population in each pixel *i*, and $C_i$ is the $PM_{2.5}$ concentration in the same pixel.

(2)   Relief degree of land surface (RDLS)

RDLS refers to the difference between the highest altitude and the lowest altitude within a certain area, which can comprehensively express the altitude and surface cutting depth of the region [33]. Its calculation formula is:

$$RDLS = \frac{ALT}{1000} + \{[Max(H) - Min(H)] \times [1 - P(A)/A]\}/500 \tag{14}$$

where *ALT* indicates the average altitude (m) of the neighborhood range centered on a certain raster cell; *Max (H)-Min (H)* are the differences between the highest and lowest altitudes (m) of the neighborhood range, respectively; *P (A)* represents the flat area (km$^2$) in the neighborhood range; and *A* is the total area (km$^2$) of the neighborhood range. In this study, the change-point analysis method was used to determine that 37 km $\times$ 37 km is the neighboring area range.

(3)   Temperature–humidity index

Climate conditions are important factors that affect human activities and the HSQ. Among many factors that affect human comfort, temperature and humidity are undoubtedly relatively important [36]. This study used the temperature–humidity index proposed by Thom as the evaluation model [51], and its calculation formula is:

$$THI = T - 0.55 \times (1 - f)(T - 58)$$
$$T = 1.8t + 32 \tag{15}$$

where *THI* is the temperature–humidity index, *T* is the average Fahrenheit temperature (°F), *t* is the average temperature (°C), and *f* is the average relative humidity (%).

(4)  Hydrological index

Hydrological conditions affect not only the natural geographical environment, but also social production and population living conditions [38]. In this study, precipitation and the proportion of the water area were used to characterize the abundance and scarcity of regional water resources. These two factors reflect the regional natural water supply capacity and the water collection and catchment capacity under natural conditions, respectively [33]. The specific calculation formula for the hydrological index is:

$$WRI = \alpha P + \beta Wa \tag{16}$$

where *WRI* is the hydrological index; *P* and *Wa* are the standardized annual precipitation and standardized water area obtained by using range standardization; and $\alpha$ and $\beta$ represent the weights of the standardized annual precipitation and standardized water area, respectively.

(5)  Ground cover index

This study referred to the ecological environment assessment standard of the State Environmental Protection Administration and integrated the basic situation of the regional land cover [39]. The ground cover index model is constructed by using land use types and the *NDVI*, which is defined as follows:

$$LCI = NDVI \times LT_i \tag{17}$$

where *LCI* is the ground cover index; *NDVI* is the normalized vegetation index; and $LT_i$ is calculated mainly with reference to the weights listed in the Technical Guidelines for the Work with National Population Development Functional Areas [18].

(6)  Construction of the *HSQ* model

The *HSQ* is composed of five indicators, including topography, climate, hydrology, PER-PM$_{2.5}$, and ground cover, which represent the regional *HSQ*. To improve the comparability of the indicators, they must be standardized. The equation of the comprehensive evaluation model of *HSQ* is as follows:

$$HSQ = \alpha_1 \times NRDLS + \alpha_2 \times NTHI + \alpha_3 \times NWRT + \alpha_4 \times NLCI + \alpha_5 \times NR_i \tag{18}$$

where *NRDLS*, *NTHI*, *NWRI*, *NLCI*, and $NR_i$ refer to the topographic relief, temperature and humidity index, hydrological index, ground cover index, and PER-PM$_{2.5}$ after range standardization, respectively. The weights of $\alpha_1$, $\alpha_2$, $\alpha_3$, $\alpha_4$, and $\alpha_5$ were calculated by the Entropy-TOPSIS and grey relational degree model.

2.4.3. Time-Series Analysis Method and Geographical Detector

(1)  Time-series analysis method

By using the Theil–Sen median method to analyze the PER-PM$_{2.5}$, the interannual variation trend of population exposure risk in the study area can be revealed [52]. In order to detect the trend inflection points of each pixel on a regional scale, this paper uses the Mann–Kendall (MK) nonparametric significance test method [53,54]. This method can

evaluate the monotonicity of the change trend of PER-PM$_{2.5}$ from a statistical perspective, with a high confidence level ($\alpha$ = 0.05). At the same time, this article uses the Hurst index to determine the future trend of PER-PM$_{2.5}$ to provide a reference for the future migration of local population and industries [55].

(2)　Optimal-parameters-based geographic detector (OPGD)

In order to explore which of the secondary indicators are the key factors affecting the HSQ, this study selected seven relevant variables affecting the HSQ, including "temperature" (TMP), "precipitation" (PRE), "PM$_{2.5}$", "land use and cover change" (LUCC), "NDVI", "relative humidity" (RH), and "DEM". At the same time, in order to detect the geographical spatial differentiation of things or phenomena and to reveal the magnitude of their underlying driving forces, this article uses GeoDetector to identify the key influencing factors [56,57]. According to this approach, an environmental variable is decisive for the change and development of a geographic phenomenon if it is significantly consistent with the spatial variability of that phenomenon [56,58]. In this study, OPGD was used to determine the optimal scale of spatially stratified heterogeneity [58].

The factor detector was used to assess the driving force of various factors on the HSQ of the MSBBC by using a q-statistic:

$$q = 1 - \frac{\sum_{h=1}^{L} N_h \sigma_h^2}{N \sigma^2} \qquad (19)$$

where $q$ is the explanatory power of the impact factors related to the HSQ, with a range of [0, 1]. The larger the $q$ value, the stronger the explanatory power of the impact factors on the HSQ; h is the stratification of variable $Y$ or factor $X$; $N_h$ and $N$ are the unit numbers of layer $h$ and the entire area, respectively; and $\sigma_h^2$ and $\sigma^2$ are the variance of the HSQ in the h division and the entire region, respectively.

The interaction detector is used to analyze the impact of interactions between different factors on the HSQ, i.e., to evaluate the joint and interactive independent effects between the two influencing factors $X_m$ and $X_n$ [58]. This is the biggest advantage of the GeoDetector model compared with other spatial statistical models.

## 3. Results

### 3.1. Characteristics of the Main Indicators of HSQ

3.1.1. Spatial–temporal Evolution Characteristics of the PER-PM$_{2.5}$

According to existing research, the PER-PM$_{2.5}$ can be divided into six levels to more intuitively observe the degree of threat to human health caused by the distribution of PM$_{2.5}$ pollution in different regions [50]. The results showed that during the study period, the spatial distribution of the PER-PM$_{2.5}$ in the MSBBC was generally similar, that is, most areas were at low risk, but the risk level in some areas were always high (Figure 3a–e). However, the PER-PM$_{2.5}$ showed remarkable differences on both sides of the Hu Line [59]. The risk level in most areas east of the line was at a relatively low level [59]. Regions with high risk levels were mainly distributed in urban agglomerations, especially in VI and IX—most areas were at high risk. These areas exhibit a concentration of population and industry. During industrial production, energy combustion, vehicle emissions, and so on make the PM$_{2.5}$ value higher, and these areas are densely populated, making the risk level higher. The PER-PM$_{2.5}$ in the area west of the Hu Line was generally low. Most of them are alpine or desert areas with sparse population and industries. Their PER value remained at a considerably low level.

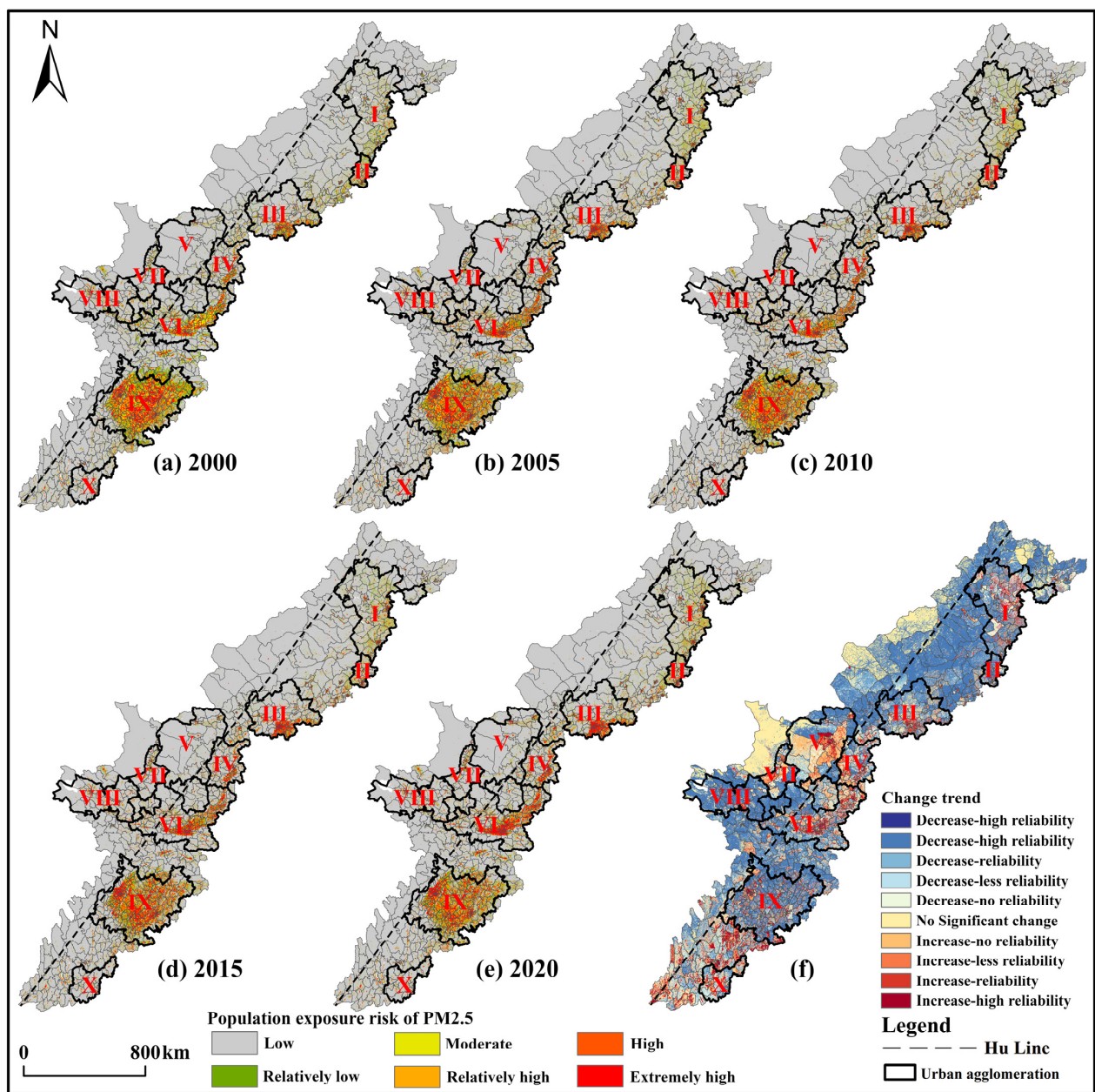

**Figure 3.** Spatial distribution (**a**–**e**) and change trend (**f**) of PER-PM$_{2.5}$ in the MSBBC from 2000 to 2020.

In general, the change trend of the PER-PM$_{2.5}$ in the MSBBC from 2000 to 2020 (Figure 3f) was mainly a reduction (71.67% of the area). The exposure risk of urban agglomerations such as Southwest, Middle East, and I mainly increased (28.01%), but the increasing rate was small, only 0–0.1. The area with no significant change (10.34%) was found in the Inner Mongolia Plateau, where the population is sparse and the industry and commerce are underdeveloped [60]. These areas are sparsely traveled and rarely distributed in industry, and most of the fine particles are seasonal dust, so no significant change occurs throughout the year. The change in the PER-PM$_{2.5}$ in the area east of the Hu Line was more dramatic, with a change rate of 0.034 μg/m$^3$·a and an overall increasing trend. The change in exposure risk in the area west of the Hu Line was mainly a non-significant change and an extremely significant decrease, with a change rate of 0.006 μg/m$^3$·a. In terms of significance level, the type of change trend in the MSBBC was mainly that of the extremely significant decline, accounting for 45.97%, followed by the non-significant decreasing zone, accounting for 13.88%. The difference between the non-significant decreasing zone and the increasing

zone (9.39%) was not significant, indicating that the PER-PM$_{2.5}$ in most of the MSBBC has been alleviated, but the problem remains serious in some areas [61]. The high exposure risk in these areas will directly affect the HSQ, thereby exacerbating the current environmental threat to human health. Therefore, it is necessary to include it as an indicator in the HSQ system for quantitative inversion.

### 3.1.2. Results of Other Indicators

The spatial difference in the RDLS in the MSBBC was obvious (Figure 4a). The RDLS was lower in most regions but high in some regions (mainly in the southwest). The RDLS of IX was significantly lower than that of the rest of the surrounding areas, resulting in a basin topography [11]. In general, the THI decreased with increasing latitude (Figure 4b). The THI in the southwest was the highest, whereas that of the Qinghai–Tibet Plateau and the northeastern border areas were lower. In addition, the THI of mountain areas at the same latitude was significantly lower than in plain areas [18]. The WRI was clearly bounded by the Hu Line in space (Figure 4c). High values were concentrated in the IX and southwestern border areas. The Inner Mongolia region west of the Hu Line was a low-value concentration area for the WRI. The WRI showed a gradually increasing trend from west to east. The spatial distribution of the LCI is similar to that of the WRI (Figure 4d). The central region in the west of the Hu Line mainly showed a low LCI. The LCI in the northeast and southwest of the east was higher because the local vegetation there is dense, whereas the land type in the western region is mainly grassland or desert, resulting in the difference in the LCI between the east and the west.

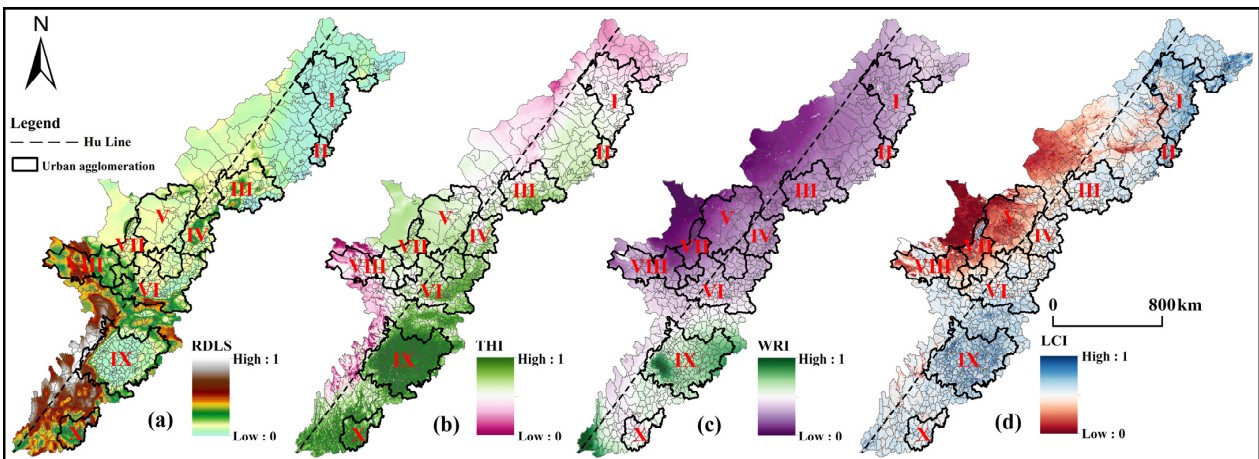

**Figure 4.** Multi-year mean spatial distribution characteristics of other indicators of HSQ in the MSBBC (**a**–**d**).

### 3.2. Spatial–Temporal Distribution of HSQ in the MSBBC

The HSQ in the MSBBC from 2000 to 2020 showed distinct fluctuations in three stages (up–down–up) (Figure 5a–e). From 2000 to 2010, the HSQ of all counties (cities) in the study area increased from 0.46 at the beginning to 0.47 in 2010. (Figure 5a–c). Although the overall regional growth rate did not rise much, the local growth rate (mainly in the southwestern cities with a relatively developed economy) was large. Among them, the HSQ in 398 counties (cities) showed a remarkable growth trend (growth rate > 1), accounting for 45.74% [62]. From 2010 to 2015, the HSQ showed a rapid decline trend, and the overall HSQ dropped to 0.43 (Figure 5c,d). At the same time, the decline rate in the HSQ in the western region was more rapid than in the eastern part. By 2020, the HSQ in the MSBBC had increased slowly again (Figure 5e). Some regions (such as Beijing, Chengdu, Guanzhong, and other metropolises) had experienced drastic changes in the HSQ due to the increase in the local PER-PM$_{2.5}$, which led to the overall decline of the local HSQ to a lower level.

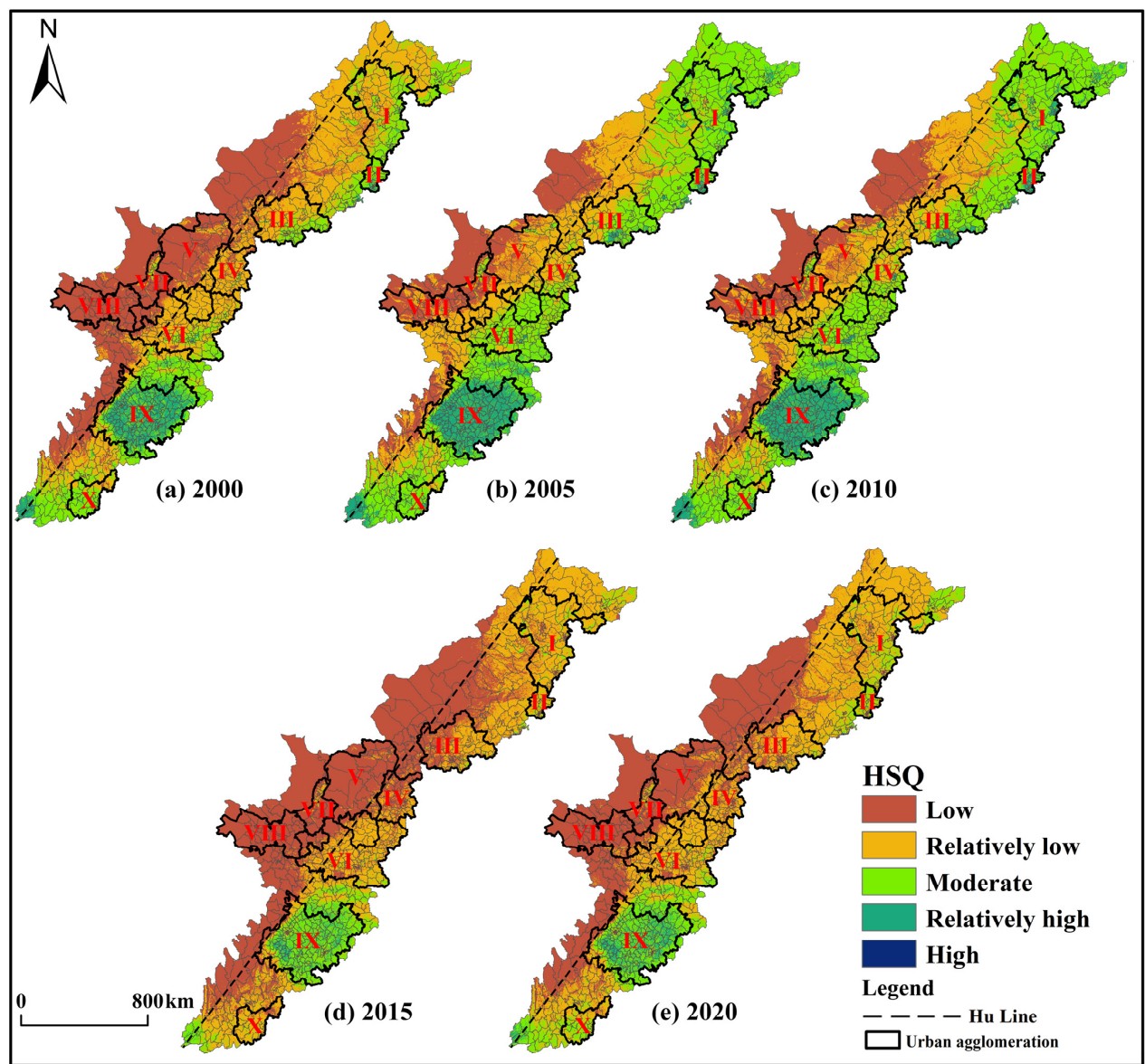

**Figure 5.** Spatial distribution characteristics of HSQ in the MSBBC from 2000 to 2020 (**a–e**).

In terms of space, the HSQ decreased from the southeast to the northwest. Topographically, the HSQ showed a trend of increasing from the mountains and plateaus to the plains and valleys. The western part of the MSBBC with higher altitudes (average altitude of 1801 m) was mainly the low-value area of the HSQ [25]. The HSQ in the Tibetan Plateau region was the lowest. The northwestern region was a sub-low point. The highest values of HSQ were distributed in the Sichuan Basin and Yunnan Province. Overall, the HSQ of the MSBBC was clearly bounded by the Hu Line. At the same time, the HSQ retained the characteristic of locally high values. The HSQ in the east and west of the Hu Line was 0.46 and 0.434, respectively. Highly habitable areas were found in the local area (IX), which has an area of 132,500 km$^2$, almost covering the entire IX area. Despite the harsh natural conditions in the west, these areas still had certain advantages in terms of air quality, making these areas more livable. In the megalopolitan areas of urban agglomeration, the HSQ was characterized as low to high level. Higher levels of PER-PM$_{2.5}$ resulted in a lower HSQ in metropolitan centers. In the suburbs, the HSQ increased. When moving away from the city at a certain distance, HSQ levels started to decrease.

### 3.3. Identifying Key Factors of HSQ Based on the OPGD

3.3.1. Driving Effect of Various Factors on HSQ

According to the factor detector results, although the driving forces of various factors on the HSQ varied year by year, the overall explanatory force for the spatial heterogeneity of the HSQ was as follows: PRE > TMP > NDVI > LUCC > DEM > $PM_{2.5}$ > RH (Figure 6a). During the study period, the interpretative power of meteorological parameters (temperature, precipitation) and the NDVI toward the HSQ was relatively stable, and their values were above 0.35 [25]. In addition, the q values of all indicators were less than 0.001, indicating that each factor had strong statistical significance, and the model had statistical significance. In 2010, the $PM_{2.5}$ pollution in the MSBBC was more serious, and $PM_{2.5}$ became one of the main driving forces of HSQ (q = 0.3). An increase in air pollutants will directly lead to the deterioration of the quality of the living environment. By 2020, the driving force of $PM_{2.5}$ pollution on the HSQ would be reduced to 0.1. Due to the vigorous development of clean industries and environmental governance by the local government, the air quality in various regions has rapidly improved, and the driving forces of $PM_{2.5}$ pollution have gradually decreased, and are even no longer the main factors limiting the gap in the HSQ across regions [63]. Therefore, in the long run, meteorological parameters (temperature, precipitation) and NDVI will be the main driving forces of HSQ.

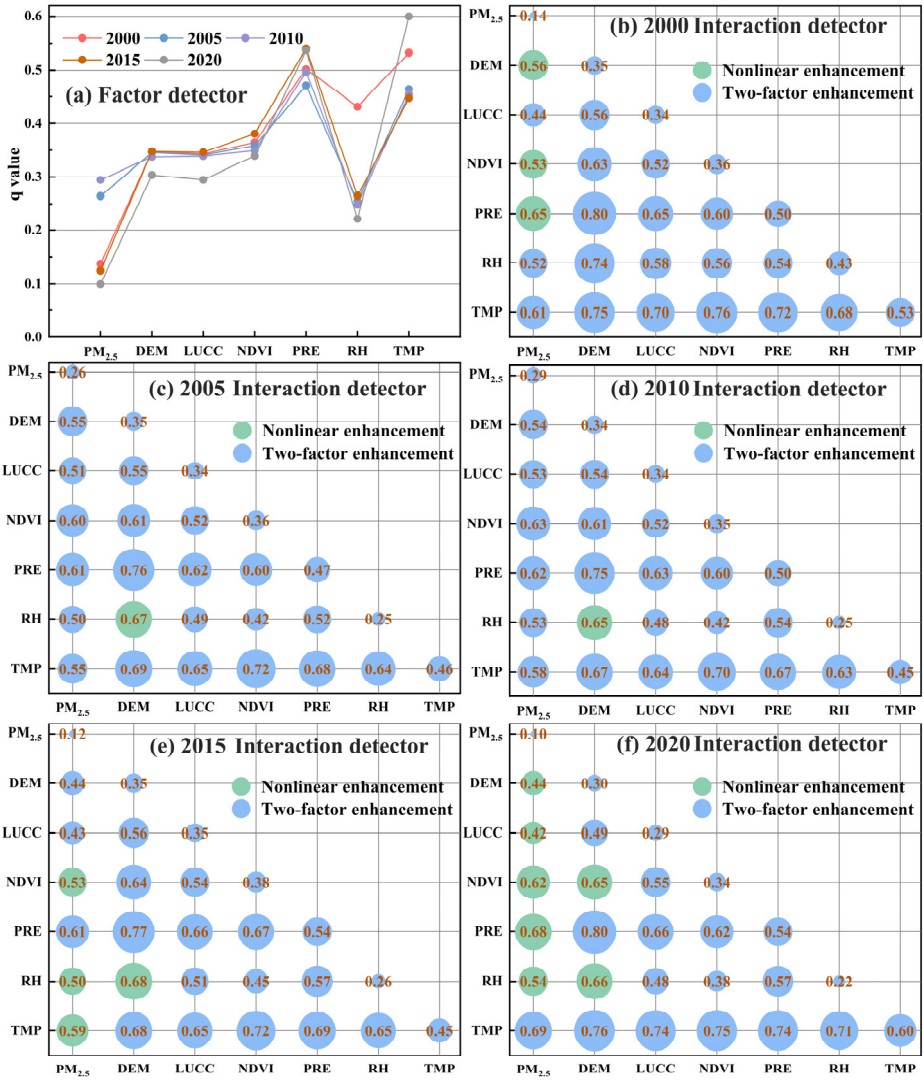

**Figure 6.** Factor detection (**a**) and interactive detection results (**b**–**f**) of factors affecting HSQ from 2000 to 2020.

### 3.3.2. Interaction of Various Factors on the HSQ

According to the results of the interaction detection, the interaction of two factors would increase the explanatory power of HSQ (Figure 6b–f). The types of interaction included two kinds of enhancement: two-factor enhancement and nonlinear enhancement, indicating that the HSQ in the MSBBC was not the result of a single factor but of a combination. In general, the interaction value of TMP ∩ NDVI was greater than 0.7 during the 21 years. Therefore, the comprehensive interaction between the TMP and the NDVI had a significant impact on the HSQ. In recent years, the changes in HSQ in the MSBBC were mainly caused by climate and land cover, rather than economy and air quality. From 2000 to 2020, the interaction value of TMP and PRE with each factor was large. The strongest interaction between the meteorological parameters and other factors indicated that the temperature and humidity index and the hydrological index were the main indicators affecting the HSQ. The interactions of the NDVI and the LUCC with each factor was smaller than that of meteorological parameters, but they cannot be ignored. Climate is difficult to change, but improving vegetation cover and land cover is a better way to improve the HSQ.

## 4. Discussion

### 4.1. Changes in HSQ in the MSBBC

From 2000 to 2010, the HSQ in most areas of the MSBBC was rising, but the overall level was not ideal. In 2010, 274 counties (cities) in the MSBBC, accounting for 31.11% of the total number of counties (cities), were classified as critically suitable and generally suitable for HSQ (Figure 7a). Before 2010, these two categories were mainly distributed in the west of the Hu Line. However, after 2010, the critical and general quality zones began spreading eastward [11,62], similar to the results found by Cong et al. However, after 2010, the change in HSQ varied from their research. From 2010 to 2015, the HSQ of the MSBBC was in a declining stage (Figure 7b). From 2015 to 2020, the HSQ began to rise again. In 2020, 667 counties (cities) in the MSBBC were critical and generally suitable with regard to their HSQ, accounting for 75.71% of the total number of counties (cities). Interestingly, the change in HSQ in the MSBBC during the study period was similar to the temperature change (Figure 7b). Although the impact of climate change on HSQ has long been proposed, these studies were limited to small areas [25]. This study complements the understanding of the impact of climate change on HSQ in a large-scale context with complex terrain. In general, the HSQ presented a spatial pattern of "high in the east and low in the west" and "high in the south and low in the north" [64], which further confirms the work conducted by Xu et al. They concluded that before 2010, the western part of the Hu Line had a single industrial development pattern and extensive industry, and the pursuit of socioeconomic development brought about insufficient air pollution control and indiscriminate deforestation, which resulted in a poorer living environment in the west [11]. Luo et al. believed that the HSQ in the east was better than that in the west as a whole [65]. Specifically, the natural environment system in the east was better than that in the west, especially in terms of the greening level and land use. This finding is consistent with the analysis results of the THI and LCI in this study. In short, if we want to improve the overall HSQ in the MSBBC, the key is to solve the problems faced by unsuitable and critical suitable areas.

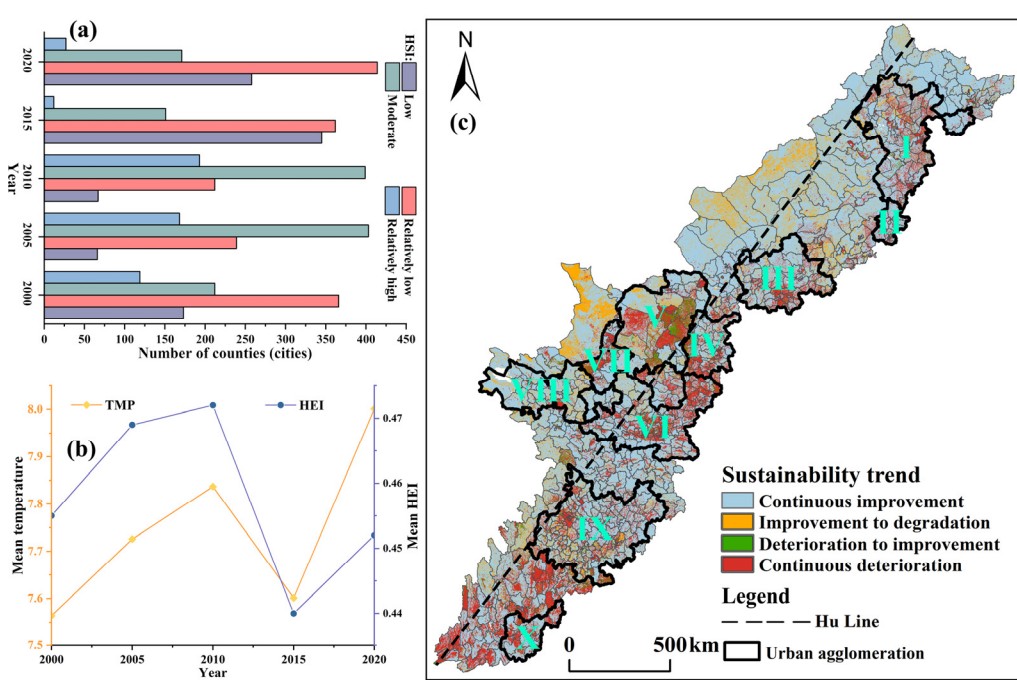

**Figure 7.** (**a**) Number of counties (cities) with different HSQ categories in the MSBBC; (**b**) change curve of HSQ and temperature; and (**c**) sustainability trend of PER-PM$_{2.5}$.

### 4.2. Impact of PER-PM$_{2.5}$ on the HSQ

During the period from 2010 to 2015, the declining trend in the HSQ in the MSBBC was not optimistic, but the HSQ was significantly improved from 2015 to 2020. Considering the development of China's national conditions, the Chinese government has issued a series of laws, regulations and standards since 2015, especially for heavy industrial cities and regions [8]. This move reduces the concentration of PM$_{2.5}$ to a certain extent, thus optimizing the HSQ at this stage [6]. At the same time, it also further supports the proposal of Zhang et al., that is, that air pollution affects the harmonious relationship among humans, the land, and human health, and that the PER-PM$_{2.5}$ has the potential to deteriorate the HSQ [7]. According to the findings of this study, the PER-PM$_{2.5}$ had a strong influence effect in the metropolitan area of the urban agglomeration but minimal impact in areas with an underdeveloped economy and population, that is, the spatial distribution of the HSQ and PER-PM$_{2.5}$ had spatial consistency [66]. This finding is contrary to research by some scholars who believe that urban agglomeration has the advantages of a convenient life, comprehensive social governance, and better economic development prospects, and that their HSQ is better than that of the surrounding areas [67]. This outcome is due to the addition of the PER-PM$_{2.5}$ as an indicator of HSQ in this study, which causes the HSQ of the metropolis in urban agglomeration to be generally lower than that of the surrounding areas [59]. Therefore, environmental supervision and pollution control should be strengthened in areas where the PER-PM$_{2.5}$ is deteriorating (Figure 7c). At the same time, it is necessary to improve the regional layout of the population distribution to avoid the existing population approaching or exceeding the ecological carrying capacity of urban agglomeration [67].

### 4.3. Analysis of Driving Factors for HSQ in the MSBBC

HSQ is the product of the interaction of various factors [68]. In factor detection, the importance of PM$_{2.5}$ at the early stage is difficult to ignore, whereas the driving force of these factors at the later stages decreased significantly. The findings of this study are attributed to the fact that the PM$_{2.5}$ pollution in the MSBBC was severe before 2010, and there was a large disparity in PM$_{2.5}$ concentrations in different regions, leading to PM$_{2.5}$ becoming one of the main driving forces of the HSQ [26]. Further analysis of the interannual

variation in $PM_{2.5}$ reveals that, since 2010, $PM_{2.5}$ pollution in the MSBBC has gradually decreased. This is mainly due to the support of government policies and the inclination of resources [69]. By 2020, $PM_{2.5}$ concentrations in almost all counties (cities) in the study area would have reached Grand II [70]. The rapid narrowing of the gap in $PM_{2.5}$ concentrations across regions has made it no longer a major driving force for HSQ.

Furthermore, this study noted that the interaction value between temperature and other factors is the largest, followed by precipitation, the DEM, and the NDVI. Based on Huo et al., the possible reason is that the DEM modulates changes in precipitation, temperature, and relative humidity, which, in turn, affect the dynamics of the vegetation [40]. Humans will choose appropriate production and living practices to change the land cover according to the natural environment, which will cause a change in temperature and humidity conditions again [71]. As pointed out by them, the return of cropland to forest and grass can enhance the retention of rainwater and maintain air humidity, etc. [71]. This finding suggests that the interaction of various driving factors can enhance the independent driving role of the HSQ. As a major driving factor of this study, temperature and humidity conditions have a strong impact on the HSQ [36]. By studying the sustainable development of the habitat environment, Wang et al. argued that development strategies should be designed in accordance with the characteristics of different regions [28]. This study suggests that the government can selectively adjust some driving factors according to local characteristics to improve the HSQ from multiple dimensions [2].

## 5. Conclusions

This study constructed a multi-dimensional evaluation framework that covers factors such as terrain conditions, climate conditions, hydrological conditions, ground cover conditions and air pollution exposure risk to evaluate the evolution of the HSQ and PER-$PM_{2.5}$ in the MSBBC from 2000 to 2020. At the same time, we utilized the OPGD to conduct in-depth research on the driving factors of the HSQ in this region. Results show the following: (1) The PER-$PM_{2.5}$ in the MSBBC was significantly different on both sides of the Hu Line. The degree of risk was positively correlated with the economy and population size. Although the annual average PER-$PM_{2.5}$ showed a decreasing trend, small and medium-sized cities still faced significant challenges in increasing their HSQ. (2) In terms of temporal changes, the HSQ in the MSBBC went through three stages of rising, falling, and rising. Among them, the HSQ in the eastern region of the Hu Line increased significantly, whereas that in the western region declined. In terms of spatial changes, the HSQ decreased gradually from east to west. (3) In terms of driving force, HSQ was more affected by meteorological factors (temperature, precipitation) than by other factors. In addition, with air quality environments, $PM_{2.5}$ was the main driving force for HSQ in the past. These elements share a mutually reinforcing relationship; therefore, enhancing vegetation coverage and land-use types can comprehensively improve the HSQ, particularly in economically developed areas with good air quality.

This study not only provides a more effective framework for evaluating human settlements, but also introduces a new perspective to explore the role of PER-$PM_{2.5}$ in HSQ. As a scientific way to discover the driving forces behind geographical phenomena, OPGD can find interesting phenomena from multiple dimensions, such as the driving force and interaction of meteorological parameters and $PM_{2.5}$, and it proposes optimization suggestions. However, certain practical reasons limit the depth of this study. First, given the lack of historical gridded data, there has been an inadequate selection of indicators and a failure to comprehensively consider the HSQ in terms of living convenience, public service development, and other aspects. Second, this study mainly considered objective environmental factors, without giving due consideration to the important aspect of human subjective feelings. Therefore, extra attention must be paid to social benefits, residential satisfaction, and material development in future research, and livable environment evaluation indicators suitable for different groups of people must be developed.

**Author Contributions:** Q.Y.: Conceptualization, Methodology, Resources, Supervision, Project administration, and Funding acquisition. J.F.: Conceptualization, Methodology, Software, and Writing—review and editing. J.M.: Methodology, Software, and Data curation. J.N.: Methodology and Funding acquisition. P.W., X.W., R.C. and Q.W.: Methodology and Resources. All authors have read and agreed to the published version of the manuscript.

**Funding:** This study was supported by the Strategic Priority Research Program of the Chinese Academy of Sciences (NO. XDA19030500), the Youth Foundation of Social Science and Humanity, China Ministry of Education (NO. 22YJCZH130), the Practice and Innovation training program for college students (NO. 202110298029Z), and the Key Laboratory for Land Satellite Remote Sensing Applications, Ministry of Natural Resources of the People's Republic of China (No. KLSMNR-G202209).

**Institutional Review Board Statement:** Not applicable.

**Informed Consent Statement:** Not applicable.

**Data Availability Statement:** The data used in this study were obtained from the following platforms: Geospatial Data Cloud for the digital elevation data (http://www. gscloud.cn (accessed on 25 October 2022)), China High Air Pollution for the $PM_{2.5}$ (https://weijing-rs.github.io/ product.html (accessed on 5 November 2022)), LandScan for the population (https://andscan.ornl.gov (accessed on 28 November 2022)), Notion Earth System Science Data Center for the temperature, relative humidity, precipitation, water area, and land cover data (http://www.geodata.cn (accessed on 2 December 2022)), National Ecosystem Science Data Center for the NDVI (http://www.nesdc.org.cn (accessed on 12 November 2022)).

**Acknowledgments:** We acknowledge the data support from "National Earth System Science Data Center, National Science & Technology Infrastructure of China. (http://www.geodata.cn (accessed on 12 November 2022))".

**Conflicts of Interest:** The authors declare no conflict of interest.

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
