# Peer review of "Assessment of Human Settlement Quality Based on the Population Exposure Risk to PM2.5 Pollution in the Mid-Spine Belt of Beautiful China"

_sustainability, doi:10.3390/su151914648_

Round 1

Reviewer 1 Report

Please read carefully the attached pdf document and consider/revise and correct all findings.

English is excellent in this article. Very few corrections found and detailed in the attached pdf file.

Reviewer 2 Report

The present manuscript is well-organized on the topic. After some editorial correction, it may be accepted for publication.

The title of the manuscript may be modified as "Assessment of human settlements quality based on population 2 exposure risk to PM2.5 pollution over the Mid- 3 Spine Belt of China"

The main outcome of the study should be highlighted in the abstract.

The definition of the problems needs to be elaborated in the Introduction section of the manuscript. This section is poorly written. It needs to be upgraded with recent literature and references.

Material and Methods as well as Results sections are reasonably well explained. In the Results section authors are suggested to add some more references in support of their statements/results.

The conclusion section needs to be shortened. Authors are suggested to provide the important outcome of the study only.

Reviewer 3 Report

The paper aims to construct an indicator for evaluating human exposure with respect to PM2.5 pollution. While the theme is interesting, the paper contains certain mistakes that currently render it unsuitable for publication in the Sustainability journal. A thorough review could potentially lead to modifications that address these issues.

In the Abstract, the author fails to clarify the paper's objectives and the contributions that have been made.

In my opinion, one of the key factors necessary to meet the substantial criteria for approval is a comprehensive review of the existing literature. The references cited in the Introduction section are limited to Chinese papers, and the absence of other articles with similar research undermines the paper's relevance to the issue highlighted by the authors.

Why was the chosen study location selected for development?

Section 2.2: A more concerted effort is needed to provide a clearer explanation of how the index system was formulated. Is there support for this in the existing literature?

Numerous methods and methodological approaches are available in the literature. Given this, why was TOPSIS specifically chosen?

Round 2

Reviewer 3 Report

there no comments